# Molecular identification and antimicrobial potential of endophytic fungi against some grapevine pathogens

**Lava H. Nashat[1], Raed A. Haleem[2]\*, Shayma H. Ali[1]**

**1** Department of Biology, College of Sciences, University of Duhok, Duhok, Duhok Province, Kurdistan Region, Iraq, **2** Department of Plant Protection, College of Agricultura Engineering Sciences, University of Duhok, Duhok, Duhok Province, Kurdistan Region, Iraq

\* raed.haleem@uod.ac

**Data Availability Statement:** All the data used in this paper is within the text.

**Funding:** The author(s) received no specific funding for this work.

## Abstract

Endophytic fungi are microorganisms that, exhibiting within the plant tissues without causing any apparent harm to the host, establish a symbiotic relationship with plants. Host plants provide endophytic fungi with essential nutrients and a protected environment. In exchange, the fungi can enhance the plant's ability to acquire nutrients. They can also play a crucial role in increasing the host plant's tolerance to various abiotic and biotic stresses. Endophytic fungi can produce a wide range of bioactive compounds, some similar to those found in the host plant. In Iraq's Duhok province of the Kurdistan region, the plant species *Vitis vinifera* has been explored as a habitat for diverse endophytic microorganisms across various ecological environments. During the period from 2021 to 2022, a total of 600 samples were collected from four distinct locations: Bagera, Besfke, Barebhar, and Atrush. From these samples, twelve endophytic fungal species were isolated, including *Aspergillus flavipes*, *Botryosphaeria dothidea*, *Fusarium oxysporum*, *Fusarium ruscicol*, *Fusarium venenatum*, *Chaetomium globosum*, *Clonostachys rosea*, *Mucor racemosus*, *Penicillium glabrum*, *Aspergillus terreus*, *Aspergillus nidulans*, and *Aspergillus niger*, *Alternaria alternata*, *Paecilomyces maximus*, *Curvularia buchloes*. These fungi were introduced for their potential as biocontrol agents against grapevine trunk diseases and grape rotting fungi, which pose significant risks to grapevine health and productivity. *Penicilium radiatolobatum*, *Botrysphaeria dothidea*, *Fusarium ruscicola*, *Fusarium venenatum*, and *Paecilomyces maximus* represented the first record as endophytes on grapevine in Iraq. Based on ITS and SSU sequencing, molecular identification confirmed these fungi's presence with sequence identities ranging from 99% to 100%. Phylogenetic analysis revealed that these endophytes could be categorized into five main clusters (A, B, C, D, and E), showing high intra-group similarity. Utilizing the Dual Culture method, the endophyte *Paecilomyces maximus* demonstrated a 70.83% inhibition rate against *Ilyonectria destructans*. In the Food Poisoning method, *A. flavipes* and *P. maximus* emerged as the most effective inhibitors of *Ilyonectria destructans*, whereas *A. terreus*, *M. racemosus*, and *P. maximus* achieved complete inhibition (100%) of *Botrytis cinerea*. Additionally, *M. racemosus* was identified as the most effective biocontrol agent against *Neoscytalidium dimidiatum*. In conclusion, the study emphasizes the potential of endophytic fungi from *Vitis vinifera* as effective biocontrol agents against grapevine

**Competing interests:** The authors have declared that no competing interests exist.

diseases, highlighting their role in sustainable vineyard management. These findings lead to further exploration and implementation of these fungi-inserted pest management strategies.

## Introduction

Endophytic fungi engage in complex interactions with their host plants, often providing mutual benefits such as enhanced growth, stress tolerance, and disease resistance. In the context of grapevines, endophytic fungi are not only integral to plant health but also hold potential for drug discovery and improving plant resilience against various stresses like disease, drought, and salinity [1]. Recent studies have demonstrated that fungal endophytes can colonize plant tissues through the development of hyphae, establishing communities within specific ecological niches [2, 3]. This colonization can result in various benefits for the host, including enhanced nitrogen fixation, protection against pathogens, and increased resilience to environmental stressors [4]. Grapevine cultivation is a significant agricultural practice worldwide, including in the Duhok Province of the Kurdistan Region. Most previous studies have focused primarily on the endophytic fungal diversity in cultivated grapevines, with less emphasis on wild grape species [5]. A comparative study of these grapevine types could yield valuable insights into the domestication process, host specificity, distribution patterns, and beneficial impacts of endophytes, as well as elucidate the diversity of endophytic fungal species.

In Duhok Province, grapevines face numerous biotic and abiotic challenges as in other vineyard regions. Adverse environmental conditions can significantly affect grapevine health and the quantity and quality of grape production [6]. The impact of cultivation practices on endophytic fungal populations in grapevines remains underexplored Furthermore, grapevines are susceptible to several fungal pathogens that cause significant crop losses. Historically, sodium arsenate was used to manage grapevine trunk diseases (GTDs), but its use was discontinued due to its severe toxicity to humans and the environment [7]. Consequently, there is a pressing need to explore alternative, safer methods for controlling GTDs. Recent research has highlighted the potential of endophytic fungi as biological control agents for effective disease management [8].

Previous research has documented the antagonistic effects of endophytic fungi on grapevine pathogens, including *Alternaria alternata*, *Aspergillus terrus*, *Botryosphaeria dothidea*, *Mucor racemusus*, *Paecilomyces maximosus*, and *Curvularia buchloes*. These fungi have shown promise in suppressing diseases caused by pathogens like *Cytalidium*, *Macrophomina*, *Neoscytalidium*, and *Botrytis* [5, 9].

In a study conducted by Martínez-Diz et al. [10], symptoms of black foot disease were observed in 2-year-old grapevines; the affected plants exhibited delayed budding, reduced vigor, and chlorotic leaves, with roots showing black discoloration and necrosis of the wood tissues. Managing vine foot rot caused by *Ilyonectria radicicola* can be improved by using soil amendments with fungicides and the antagonistic fungus *Trichoderma harzianum*, which may help suppress the disease [11]. In Iraq, grapevine decline caused by *Ilyonectria destructans* was frequently observed and successfully controlled in vitro using two bioagents: *Trichoderma harzianum* and *Clonostachys rosea* [12, 13]. Endophytic fungi have also shown promise as biocontrol agents against grapevine trunk diseases (GTDs) [14]. *Botrytis cinerea* is responsible for gray mold, a major disease that impacts grapevines (*Vitis vinifera* L.) and leads to considerable losses in both yield and quality globally; this pathogen can exist as a saprophyte, necrotrophy, or parasite, affecting various grapevine parts, including leaves, green shoots, rachises, flowers, bunch debris (such as calyptras, dead stamens, aborted flowers and berries, and tendrils), and

ripening berries [15]. Several commonly occurring fungi, such as *Alternaria spp.*, *Cladosporium spp.*, and *Epicoccum nigrum*, as well as basidiomycetous yeasts like *Aureobasidium pullulans*, have been isolated from grape tissues and found to act antagonistically against *Botrytis cinerea* [16]. Other studies have demonstrated that using mechanical grape harvesters can effectively reduce the incidence of Botrytis bunch rot, leading to significant cost savings [17]. Non-pathogenic microorganisms are effective biocontrol agents with minimal effects on human health and the environment. They are valuable in combating resistance strategies by managing *B. cinerea* through several mechanisms, such as competing for nutrients and space, producing antibiotics, parasitizing the pathogen, and inducing resistance in the host plant [18].

Our study sought to expand the ecological understanding of grapevine endophytic fungi, investigate their potential as biocontrol agents, and establish a basis for future research on sustainable disease management strategies, by identifying them through traditional morphological methods and molecular techniques such as DNA sequencing.

## Materials and methods

### Sample collection and study site

Samples were collected from four locations within Duhok City (Bagera and Besfke, Baribhar, and Atrush) between 2021 and 2022. A total of 600 samples were obtained, including various plant parts: trunk (ST), nodes (N), internodes (IN), roots (R), leaves (L), petioles (P), and mature fruit (MF). These samples were collected from plants cultivated using two distinct training methods: the Head training system and the Cordon training system. The collected samples were placed in sterile plastic bags and stored in a controlled environment inside a cool box. These samples were then handled within 24 hours of collection and maintained at a temperature of 4°C until use [19].

Surface disinfection is an essential step in the isolation of endophytic fungi, as it effectively removes epiphytic microorganisms that may impede the identification of true endophytes. The disinfection protocol involves several critical steps: initially, samples are washed with tap water to remove visible contaminants such as dust, debris, yeasts, and filamentous fungi [20]. Following this, the samples are submerged in 70% ethanol for 30 seconds, then treated with sodium hypochlorite (NaOCl) for 2 minutes, and subsequently immersed in 70% ethanol for 15 seconds. Finally, the samples undergo multiple rinses with sterilized, autoclaved distilled water. This comprehensive method ensures the plant materials are thoroughly disinfected and prepared for further analysis.

### Culturing of endophytic fungi

The plant samples were plated on Potato Dextrose Agar (PDA) and Malt Extract Agar (MEA) supplemented with streptomycin (100 mg/L) to inhibit endophytic bacteria [21] and incubated at 25°C for 10 days. Fungal growth was monitored daily. The isolates were grouped based on their morphological characteristics. To facilitate further identification, the hyphal tips of fungal colonies were transferred to PDA slants. This isolation method, as recommended by multiple sources [22, 23] involves collecting hyphae from the periphery of the fungal colonies and introducing them into a fresh culture medium with antibiotics. After purification, the fungal isolates were classified based on their macro- and micro-morphological characteristics [24].

### Molecular identification

**DNA extraction.** Genomic DNA was extracted with the established CTAB methods [25]. Briefly, the whole endophytic fungus was isolated from pure fungal cultures. The

**Table 1. Primer pairs used in this study.**

| Gene \loci | PCR primer Sequences 5′-3′ (forward\reverse) | Size (bp) | Annealing temp. | References |
|---|---|---|---|---|
| ITS | ITS1: TCCGTAGGTGAACCTGCGG<br>ITS4: TCCGCTTTATTGATATGC | 500 | 55˚C for 45 sec | [27] |
| SSU | NS1: GTAGTCATATGCTTGTCTC<br>NS4: CTTCCGTTCAATTCCTTTAAG | 1500 | 55˚C for 30 sec | [27] |

Erlenmeyer flasks were used to introduce endophytic fungal mold strains into 200 ml of potato dextrose broth. The flasks were incubated for ten days at a temperature of 25˚C, and cell walls of fungal mycelia were broken down by liquid nitrogen [26]. The ground fungi were subjected to CTAB protocol and purified using phenol: chloroform: isoamyl (25:24:1). Nucleic acids were precipitated by isopropanol. The precipitated nucleic acids dissolved in free nuclease water and stored at -4˚C until used. The quantity and the quality of the extracted DNA were measured by a NanoDrop 2000UV-spectrophotometer, at two distinct wavelengths 260 nm and 280 nm. The purity of DNA was determined using the absorbance ratio A260\A280.

**PCR amplification of ITS and SSU regions.** Amplification of ITS (ITS1, ITS4) and SSU (NS1, NS4) consisted of 12.5 μl of Taq PCR master mix, 1 μl of both forward and reverse primer of 10 (pmol\μl), 2 μl genomic DNA (25-50ng\μl), and 8.5 μl of sterile deionized distilled water in 25μl of the final reaction volume. Amplifications were carried out in an Eppendorf AG / USA thermal Cycler using the amplification conditions as follows: PCR steps for ITS regions: an initial denaturation cycle for 5 minutes at 95˚C, followed by 35 cycles starting with denaturation at 94˚C for 1 minute, and annealing at 55˚C for 45 sec. and extension at 72˚C for 45 sec followed by a final extension for 7 minutes at 72˚C. while the PCR conditions of SSU were as follows: initial denaturation cycle at 95˚C for 5 minutes followed by 40 cycles beginning with denaturation for 35 sec at 96˚C, annealing at 55˚C for 30 sec., extension for 1 minute at 72˚C followed by a final extension at 72˚C for 5 minutes. The amplified PCR products of ITS (500bp) and SSU (1500bp) were separated by agarose gel electrophoresis, and observed under a UV- transilluminator. The pair of primers were the sequences of primer utilized in this study as shown in Table 1.

**Sequencing of ITS and SSU regions.** Qiagen Minielute purification kit purified PCR products following the manufacturer's instructions. Macrogen Incorporation (Seoul, South Korea) sequenced the purified PCR products using an ABI3730 XL automatic DNA analyzer and the primer pair ITS1, ITS4, and SSU (NS1, NS4).

**Analysis of ITS and SSU regions and species identification.** Geneious version R 8.1 Biomatters 14, and Bio Edit version 7.2 software programs were applied to edit, analyze, trim, and verify the sequenced fragments of both forward and reverse primers for both regions and saved in FASTA format. ITS (Internal Transcribed Spacer) and SSU (Small Subunit) amplicon-sequenced fragments were compared with available sequences at NCBI (National Center for Biotechnology Information). Using the Basic Local Alignment Search Tool (BLAST) against ITS and SSU sequences of type isolates (www.ncbi.nlm.nih.gov/ BLAST, the nucleotide sequences showed ≥ 99% similarities.

**Phylogenetic analysis.** Bio Edit and MEGA software programs [28], were used for phylogenetic analysis and nucleotide sequence alignment. The Cluster W algorithm using the default parameters was employed to align sequences. The Neighbor-joining method was applied to construct a phylogenetic tree [29].

### In-vitro antagonism potential of bioagents

**Dual culture method.** Each set of mycelium plugs from endophytic and pathogenic fungi was placed on a potato dextrose agar (PDA) plate, maintaining a 4 cm distance between them. The distance from the edge of the colonies to the plugs was recorded as 0.5 cm. The endophytic fungus was placed 2 cm from one edge of the petri dish, while the pathogen was positioned on the opposite side. The plates were incubated at 25±2°C for 10 days, with three plates prepared for each isolation. Control treatments involved pure fungal cultures. Once the pathogen had fully grown in the control, the radial growth of all treatments was measured.

The percentage of inhibition of mycelial growth of the test organisms compared to the control was calculated using the formula: I % = C-T/C×100, where I represents rate inhibition, C represents the rate of controlled growth and, T represents the rate of growth in the treatment [30].

**Food Poisoning method.** The culture from each isolated bioagent was transferred and placed in a 250 mL Erlenmeyer flask containing 50 mL of Potato Dextrose Broth (PDB). The flask was then incubated in the dark at a speed of 150 rpm for 72 hours. Following the incubation period, the fungal spores in each Bioagents culture were eliminated by filtration using Whatman N°4 filter paper and a 0.45 μm. Millipore membrane, using a protocol adapted by Frighetto, [31]. Petri plates were filled with Potato Dextrose Agar (PDA) medium and 10 mL of each filtrate, each with a (50%) concentration, was applied to the plates. After solidification, a seven-millimeter culture disc of pathogenic fungus was placed in the center of a Petri plate. The mycelial diameter of the pathogenic fungus was measured using two perpendicular orientations after the pathogens were inoculated in the experimental group's Petri plate. The rate of suppression of mycelial growth in the test organism compared to the control was calculated using the formula proposed by Vincent [30].

## Results and discussion

### Prevalence of endophytic fungi

Twelve endophytic fungi were identified from healthy *Vitis vinifera* plants in four regions of Duhok province in Iraq (Bagera, Besfke, Baribhar, and Atrush). The following fungus was isolated: *Aspergillus terreus*, *Aspergillus nidulans*, *Aspergillus niger*, *Aspergillus flavipes*, *Botryosphaeria dothidea*, *Fusarium oxysporum*, *Fusarium ruscicol*, *Fusarium venenatum*, *Chaetomium globosum*, *Clonostachys rosea*, *Mucor racemosus*, and *Penicillium glabrum*. *Mucor racemosus* is the only fungus in this study that is not classified under the Ascomycota group; which belongs to the Zygomycota, with the highest occurrence in Besefki, Baribhar and Atrush regions (100%), (83.3%), and (83.3%), respectively. While *Clonostachys rosea* most occur in Bagera (100%) (Table 2). A study conducted by AL-Rifaie and Ameen [32] represents the first investigation into the family Chaetomiaceae as endophytic fungi in Basrah, Iraq, with a specific focus on vegetable samples. Ascomycota phylum consistently dominates the fungal endophyte population in *Vitis vinifera*, regardless of the host plant's geographical location. The most prevalent genera identified include *Penicillium*, *Cladosporium*, *Didymella*, *Aspergillus*, *Aureobasidium*, and *Alternaria*. *Alternaria* and *Cladosporium* are among the most abundant endophytes in *Vitis vinifera* [33].

### Molecular identification and phylogenetic analysis

The qualification and quantification results revealed the concentration of the isolates endophytic fungi ranged between (200–1300 ng/μl) with a purity value between (1.6–1.82). The sequenced results of ITS and SSU indicated a diversity of endophytic fungi present in various grapevine tissues. Specifically, seven fungi were from leaves six from inter nods, eleven from

**Table 2. The percentage frequency of endophytic fungi in four distinct areas within the Duhok province in Iraq.**

| Endophytic fungi | Location | | | |
|---|---|---|---|---|
| | Bagera | Besefke | Baribhar | Atrush |
| *Aspergillus terreus* | 33 | 83.3 | 50 | - |
| *Aspergillus nidulans* | - | 33 | 33 | - |
| *Aspergillus niger* | 66.7 | 83.3 | 66.7 | 66.7 |
| *Aspergillus flavipes* | 50 | 50 | 83.3 | - |
| *Botryosphaeria dothidea* | 33 | - | - | 66.7 |
| *Fusarium oxysporum* | 66.7 | 66.7 | 33 | 33 |
| *Fusarium ruscicol* | - | 33 | - | - |
| *Fusarium venenatum* | - | 50 | - | - |
| *Chaetomium globosum* | 33 | 33 | - | 33 |
| *Clonostachys rosea* | 100 | 16.7 | - | 50 |
| *Mucor racemosus* | 50 | 100 | 83.3 | 83.3 |
| *Penicillium glabrum* | 66.7 | 50 | 33 | 50 |
| *Alternaria alternata* | 84.3 | 56.7 | 73.1 | 56.2 |
| *Penicilium radiatolobatum* | 26.1 | 15.3 | - | - |
| *Curvularia buchloes* | 15.1 | - | 15 | - |
| *paecilomyces* maximosus | 50 | 33 | 33 | - |

nods, five from petioles, four fungi from fruits, seven fungi from stems, and eight fungi isolated from roots as shown in (Table 3). Particularly, the distribution of endophytic fungal varied significantly across different tissues of the grapevine, with colonization rates being distinctly higher in nod compared to others. While Zheng indicated the colonization rates of these fungi are significantly higher in stems compared to leaves. Additionally, fungal communities within leaf and root tissues demonstrate significant differences in composition and diversity [34]. These organisms do not harm the host but have many benefits as improving host growth, supplying the host with nutrients, etc. This finding agrees with Baron who highlighted the potential of endophytes to provide benefits to their hosts [35], The deepening of studies involving

**Table 3. Diversity of isolated fungi species within plant tissues based on ITS and SSU accession numbers.**

| Plant tissues | Isolated endophytic fungi | Similarity | ITS accession number | SSU accession number |
|---|---|---|---|---|
| Root, and inter nod | *Fusarium oxysporum* | 99.81 | PP410452 | PP342393 |
| Root, nod, leaf, petioles, and fruit | *Aspergillus terrus* | 99.82 | PP410453 | PP342394 |
| Root, and nod | *Botryosphaeria dothidea* | 99.80 | PP410462 | PP342404 |
| Root, stem, nod, inter nod, leaf, and petioles | *Alternaria alternata* | 99.81% | PP410473 | _ |
| Root, stem, inter nod, leaf, and fruit | *Mucor racemosus* | 100 | PP410456 | PP342397 |
| Root, Stem, nod, inter nod, leaf, petioles, and fruit | *Aspergillus niger* | 100 | PP410457 | PP342398 |
| Root | *Aspergillus flavipes* | 100 | PP410458 | PP342400 |
| Root, Stem, nod, leaf, petioles, and fruit | *Chaetomium globosum* | 100 | PP410459 | PP342401 |
| Stem | *Aspergillus nidulans* | 100 | PP410454 | PP342395 |
| Stem, nod, and inter nod. | *Fusarium ruscicola* | 100 | PP410461 | PP342403 |
| Stem, nod, leaf, and petioles | *Penicillium glabrum* | 100 | PP410460 | PP342402 |
| Nod and inter nod. | *Penicilium radiatolobatum* | 99.82% | PP410466 | _ |
| Nod | *Fusarium venenatum* | 100 | PP410463 | PP342405 |
| Nod | *Clonstachys rosea* | 100 | PP410455 | PP342396 |
| Nod, leaf | *Curvularia buchloes* | 100.00% | PP410464 | _ |
| stem, nod, inter nod, and Leaf | *paecilomyces* maximus | 100% | PP410469 | _ |

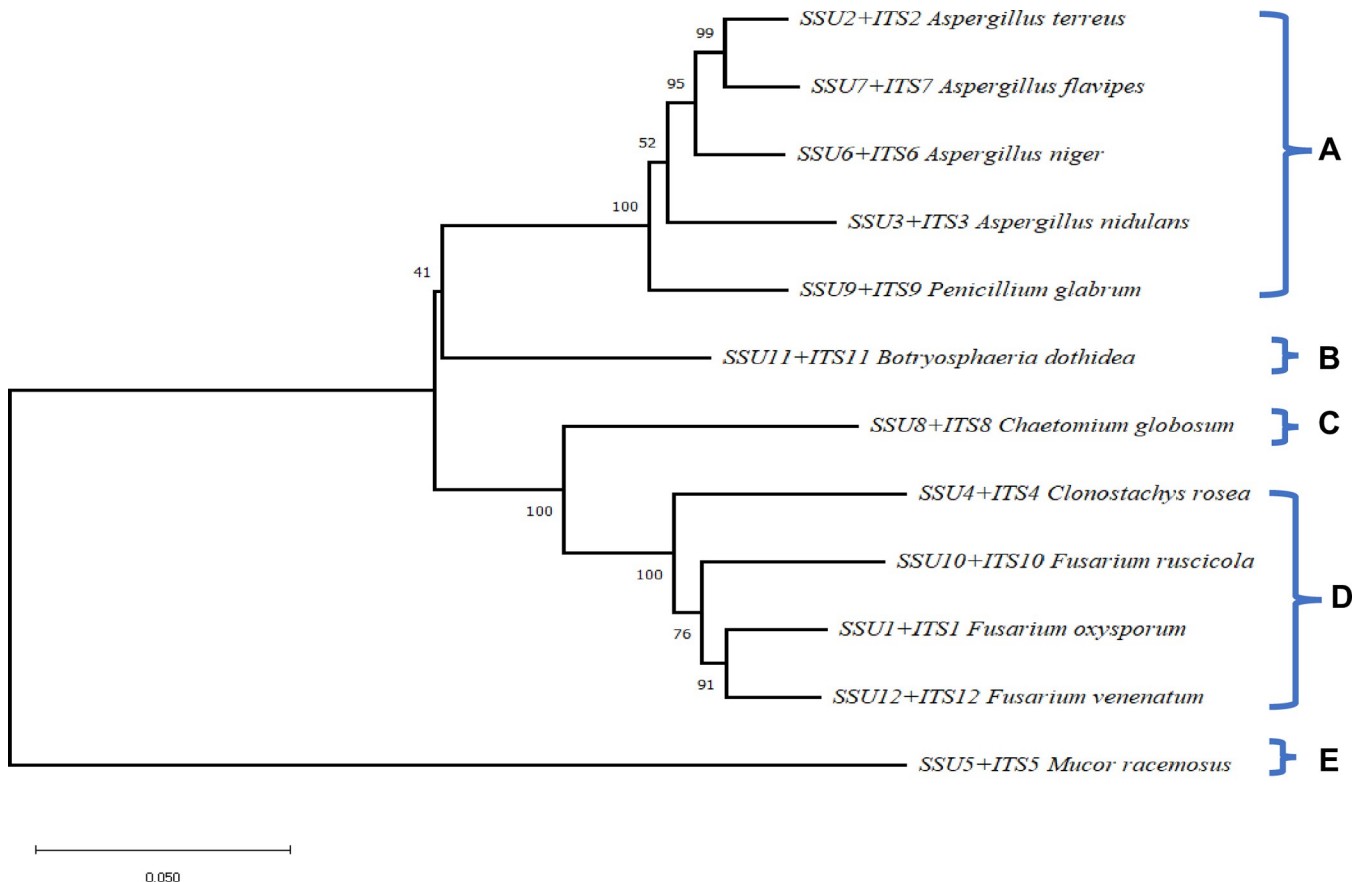

**Fig 1.**

interactions between microorganisms considered beneficial to plants and their hosts has shown that the plant genome interacts with microorganisms, which has allowed the exploration of a new aspect in the search for more sustainable agriculture [36].

Blast analysis of SSU and ITS regions resulted in seven genera of endophytic fungi (Fig 1). Different evolutionary groups are shown in the phylogenetic relationships between different fungal species, as illustrated in the trees of ITS and SSU genetic regions. Group A comprises *Aspergillus species* (*A. terreus*, *A. flavipes*, *A. niger*, and *A. nidulans*), which exhibit significant confidence in their connections with high bootstrap values indicating close relatedness. In Group B, *Aspergillus* species *and Penicillium glabrum* have a close but distinct relationship. Different from other fungi, *Botryosphaeria dothidea* is the representative of Group C. Group D includes *Clonostachys rosea*, *Chaetomium globosum*, and many *Fusarium* species (*F. rusicola*, *F. oxysporum*, *F. venenatum*). Fusarium forms a well-supported subgroup. The most distantly related species in Group E is *Mucor racemosus*, suggesting an early split from the common ancestor. This tree illustrated the diversity of fungal evolution by highlighting the closer relationships among *Aspergillus* species, the different evolutionary paths taken by *Penicillium glabrum* and *Botryosphaeria dothidea*, the diversity of *Fusarium* species, and the special evolutionary history of *Mucor racemosus*. Taken together, the phylogenetic results depict the potential of using combined full SSU and ITS sequences to distinguish between endophytic fungal species belonging to different genera. The level of the similarity of the obtained sequences ranged from 99–100%.

## Bioagent control

**Inhibition of pathogens growth using bioagents endophytes by dual culture method.**
The data shown in Table 4 demonstrate the efficacy of the endophyte *P. maximus* as a bioagent in inhibiting the growth of *Ilyonectria destructans*, using which achieves a (70.83%) inhibition rate, followed by (43.13%) with *Botrytis cinereal*, *Neoscytalidium dimidiatum*, (36.20%), then reduce to (29.27%) with *Macrophomina phaeolina*. Also, some references observed that *Paecilomycis* sp. is a cosmopolitan fungus mainly known biological control agent of several fungi and phytopathogenic bacteria [37]. Antagonists' rapid development is an important advantage in their competition for space, nutrition, and control over their hosts [38, 39] *A. alternata* had the lowest effect (7.87%) compared to other bioagents.

Fig 2 shows that *Aspergillus flavipes* had the highest bioagent effect at (46.69%), primarily through competition for nutrients and space, *Paecilomyces maximus* came in second with (44.86%), while *Mucor racemosus* ranked third, inhibiting pathogen growth by (32.03%). Another study also revealed the antagonistic effects of Aspergillus species, such as *A. niger* (isolate A10) and *A. candidus* (isolate A5). These isolates showed the highest in vitro inhibition of *Pythium ultimum* growth and demonstrated the ability to promote apple seedling growth [40]. Paecilomyces is a widespread fungus typically found in food, soil, and decaying plants species of Paecilomyces represent a novel class of biocontrol agents, known for their diverse antimicrobial properties, including parasitism, competition for nutrients, production of bioactive secondary metabolites, and induction of disease resistance [37]. Despite their extensive use in controlling pathogenic nematodes in plants [41], there is relatively limited information regarding their antifungal activity

**Food poisoning method.**   Table 5 Demonstrates that the endophytes *A. flavipes* and *P. maximus* were the most effective bioagents in inhibiting the growth of *Ilyonectria destructans*, while *A. terreus*, *M. racemosus*, and *P. maximus* highly affected *Botrytis cinerea's* growth (100% inhibition). Additionally, *M. racemosus* was found to be the most effective bioagent in inhibiting the growth of *Neoscytalidium dimidiatum*. This finding agreed with the reports by El-Sayed et al. [42] who found that *Aspergillus flavipes* acted as a potent inhibitor of the growth of several Phytophthora species. Furthermore, the crude extracellular extract from *A. flavipes* broth cultures demonstrated significant growth inhibition against various Phytophthora species. The in vitro and in vivo evaluations of *Aspergillus terreus* against *Fusarium oxysporum* in the study conducted by Alhaddad et al [43] highlighted the potential of endophytic fungi as efficient biological control agents. *Curvularia buchloes* appeared with the lowest inhibition rate among bioagent fungi. the low inhibition rate of *Curvularia buchloes* against these pathogens

**Table 4. Inhibition percentage of pathogenic fungi using Dual culture method.**

| Endophytic fungi (bioagent) | Pathogens | | | |
|---|---|---|---|---|
| | *Ilyonectria destructans* | *Botrytis cinerea* | *Macrophomuna phaseolina* | *Neoscytalidium dimidiatum* |
| *Alternaria alternata* | 62.50 a-c* | 7.87 o | 14.47 m-o | 11.77no |
| *Botrysphaera dothidae* | 50.00 c-f | 21.57 i-o | 24.07 i-n | 47.07 d-g |
| *Mucor racemosus* | 62.50 a-c | 31.37 h-k | 18 .53 j-o | 15.73 k-o |
| *Aspergillus terrus* | 58.33 a-d | 17.70 k-o | 21.50 i-o | 17.63k-o |
| *Paecilomyces maximus* | 70.83 a | 43.13 e-h | 29.27 h-m | 36.20 f-i |
| *Aspergillus flavipes* | 66.67 ab | 31.37 h-l | 35.93 f-i | 52.77 b-e |
| *Curvularia buchloes* | 58.33 a-d | 15.70 k-o | 33.37 g-j | 27.77 i-m |

*Means followed by different letters are significantly different at ≤ 0.05 according to the Duncan Multiple range test.

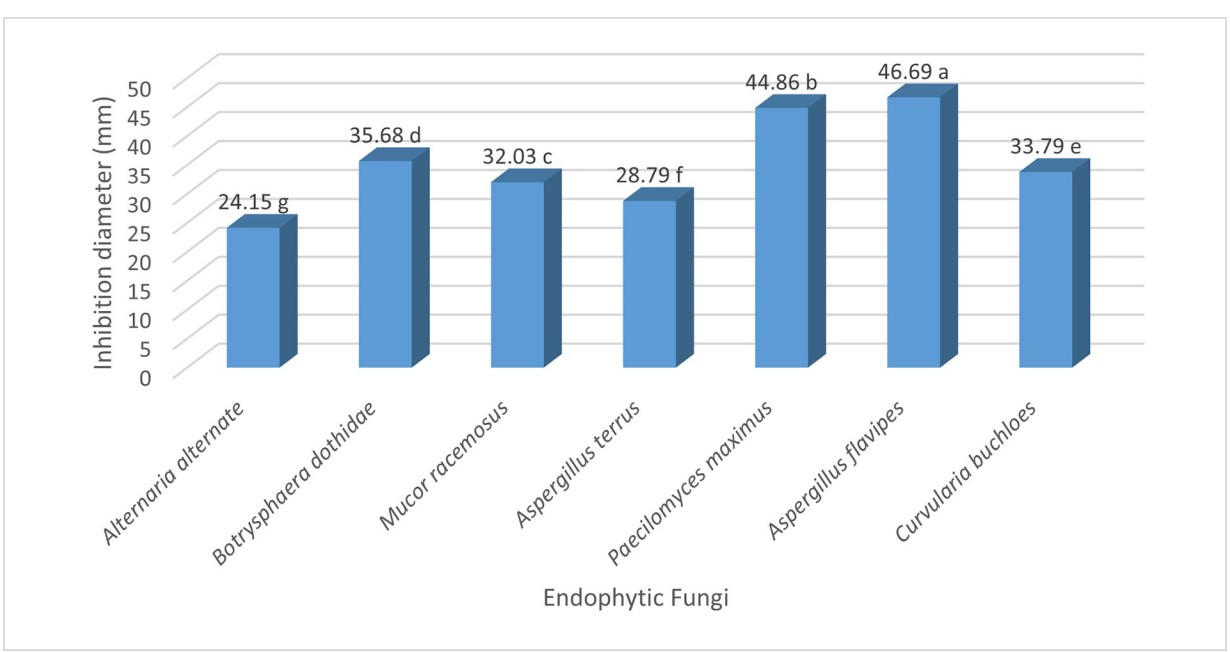

**Fig 2.**

might be influenced by a lack of the necessary mechanisms for inhibition of the germination of these pathogens (Table 5).

Endophytes from the genera *Epicoccum*, *Cladosporium*, and *Alternaria* have demonstrated their potential as biological control agents against fungi associated with esca disease [5]. Other endophytic species such as *Acremonium*, *Alternaria*, *Arthrinium*, *Ascorhizoctonia*, *Aspergillus*, *Aureobasidium*, *Bipolaris*, *Botryosphaeria*, *Botrytis*, *Chaetomium*, *Cladosporium*, *Curvularia*, *Hypoxylon*, *Lasiodiplodia*, *Mycosphaerella*, *Nigrospora*, *Penicillium*, *Phoma*, *Scopulariopsis*) have already been identifying as endophytic fungi in the earlier study that isolated from stem of grapevine, in Beijing of China [44]. Endophytic fungi have been great potential as biocontrol agents against the diseases of grapevine that caused by pathogens. Isolating and identifying endophytes from a grapevine, evaluating their antagonistic activity against major pathogens, and investigating the mechanism behind their biocontrol potential. It emphasizes the

**Table 5. Inhibition percentage of pathogenic mycelial growth treated with bioagent filtrate.**

| Endophytic fungi (bioagents) | Pathogens | | | |
|---|---|---|---|---|
| | *Ilyonectria destructans* | *Botrytis cinerea* | *Macrophomina phaseolina* | *Neoscytalidium dimidiatum* |
| *Alternaria alternata* | 33.33 e-g* | 13.77 g-i | 12.97 g-j | 58.30cd |
| *Botrysphaera dothidae* | 29.17 e-h | 15.67 g-j | 5.60 ij | 25.00 f-j |
| *Mucor racemosus* | 75.00 bc | 100.00 a | 85.20 ab | 100.00 a |
| *Aspergillus terrus* | 25.00 f-j | 100.00 a | 25.93 f-j | 19.60 f-j |
| *Paecilomyces maximus* | 100.00 a | 100.00 a | 49.27 de | 75.30 bc |
| *Aspergillus flavipes* | 100.00 a | 49.03 de | 28.13 e-i | 34.50 e-g |
| *Curvularia buchloes* | 41.67 d-f | 3.93 j | 7.40 h-j | 7.87 h-j |

*Means followed by different letters are significantly different at $\leq 0.05$ according to the Duncan Multiple range test.

*Ilyonectria destructans* showed the greatest growth but the least resistance to most antagonists, while *Botrytis cinerea* was the most affected by the antagonistic process.

importance of early detection of pathogens in vineyard nurseries that could help prevent spreading of pathogen.

## Conclusions

This study reveals the diversity and effectiveness of endophytic fungi from *Vitis vinifera* in the Duhok province-Iraq as biocontrol agents against grapevine diseases. Twelve fungal species were identified, with *Paecilomyces maximus*, *Aspergillus flavus*, and *Mucor racemosus* showing strong inhibition of major grapevine pathogens. These findings support using endophytic fungi as sustainable alternatives for disease management in vineyards, encouraging further research and practical applications in integrated pest management.

## Supporting information

**S1 Table.**
(XLSX)

**S2 Table.**
(XLSX)

## Author Contributions

**Data curation:** Lava H. Nashat, Shayma H. Ali.

**Funding acquisition:** Lava H. Nashat.

**Investigation:** Raed A. Haleem.

**Methodology:** Lava H. Nashat, Shayma H. Ali.

**Resources:** Lava H. Nashat, Shayma H. Ali.

**Software:** Shayma H. Ali.

**Supervision:** Raed A. Haleem.

**Writing – original draft:** Raed A. Haleem.

**Writing – review & editing:** Shayma H. Ali.

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
