## [Decision Letter · Decision Letter 0]

26 Aug 2024

PONE-D-24-32835Molecular Identification and Antimicrobial Potential of Endophytic Fungi Against Some Grapevine PathogensPLOS ONE

Dear Dr. Haleem,

Thank you for submitting your manuscript to PLOS ONE. After careful consideration, we feel that it has merit but does not fully meet PLOS ONE’s publication criteria as it currently stands. Therefore, we invite you to submit a revised version of the manuscript that addresses the points raised during the review process.

ACADEMIC EDITOR: The revewers have recommended a major revision. I suggest to follow closely all comment to fix all concerns that the reviewers have. ==============================

We look forward to receiving your revised manuscript.

Kind regards,

Estibaliz Sansinenea

Academic Editor

PLOS ONE

Journal Requirements:

2. In your Methods section, please provide additional information regarding the permits you obtained for the work. Please ensure you have included the full name of the authority that approved the field site access and, if no permits were required, a brief statement explaining why

The name of the colleague or the details of the professional service that edited your manuscriptA copy of your manuscript showing your changes by either highlighting them or using track changes (uploaded as a *supporting information* file)A clean copy of the edited manuscript (uploaded as the new *manuscript* file)”

4. We note that your Data Availability Statement is currently as follows: "All relevant data are within the manuscript and its Supporting Information files."

If there are ethical or legal restrictions on sharing a de-identified data set, please explain them in detail (e.g., data contain potentially sensitive information, data are owned by a third-party organization, etc.) and who has imposed them (e.g., an ethics committee). Please also provide contact information for a data access committee, ethics committee, or other institutional body to which data requests may be sent. If data are owned by a third party, please indicate how others may request data access."

Additional Editor Comments:

The revewers have recommended a major revision. I suggest to follow closely all comment to fix all concerns that the reviewers have.

Reviewers' comments:

Reviewer's Responses to Questions

Comments to the Author

1. Is the manuscript technically sound, and do the data support the conclusions?

Reviewer #1: No

Reviewer #2: Yes

2. Has the statistical analysis been performed appropriately and rigorously? 

Reviewer #1: No

Reviewer #2: Yes

3. Have the authors made all data underlying the findings in their manuscript fully available?

Reviewer #1: Yes

Reviewer #2: Yes

4. Is the manuscript presented in an intelligible fashion and written in standard English?

Reviewer #1: No

Reviewer #2: Yes

5. Review Comments to the Author

Reviewer #1: The manuscript explains isolation of endophytic fungal strains from grapevine plants from four districts of Iraq. The strains were reported as characterized through morphological and molecular tools. Further, the strains were tested for antagonistic effects against the grapevine pathogens using dual culture assay and food poisoning methods. In my opinion, the study was poorly designed with no statistical design, data analysis, poor results presentation in terms of tables and figures. The major comments are given below.

1)Line 14- 15. The endophyte P. maximus is not included in the 12 isolates, then why it was shown as the most effective inhibitor of grapevine pathogen in the manuscript. This strain was not shown in either the abstract or in the tables.

2)Some pathogenic fungi causing plant diseases were also isolated such as B.dothidea, and F.oxysporum. How these can be used as the biocontrol agents?

3)Line 45-48. Please provide more citations explaining the endophytic antagonistic effects against B.ceneria and I.destructans, two of the most destructive pathogens of grapevine.

4)Lines# 96-107. Provide details and quantities of PCR reaction mixture for both ITS and SSU.

5)Several grammatical errors were found in lines # 116- 159.

6)Table 1 needs formatting. It’s not the correct style.

7)Grammatical errors in lines# 170- 185.

8)Fig. 1. Species names must be in italics. Provide reference of NJ method.

9)Line#207. Where is P.maximus in table 1?

10)Table 3. Not the correct style. Needs formatting.

11)Fig 2 needs major revision. No horizontal and vertical titles, no error bars.

12)The results and findings have not been discussed properly in the light of previous literature. Only four references were mentioned to support the results of the present study.

Reviewer #2: Abstract

1. Line 2-5- The geographical location of the study location must be known eg. Nigeria etc

2. A brief introduction to the subject must be given in 2 lines

3. Line 9- " These fungi were "

Intro

4. Line 51- Our study sought to....."

Methods

5. More than 1 medium should have been used to plate the samples since it creates more diversity in species of fungi. A better picture is given

Results and Discussion

5. Line 210- A. alternative MUST be changed to A. alternata ......check the spelling

6. Results from this present study must be compared with similarly one works from across the globe to ascertain differences and possible sound scientific reasons given.

7.

6. PLOS authors have the option to publish the peer review history of their article (what does this mean?). If published, this will include your full peer review and any attached files.

Do you want your identity to be public for this peer review? For information about this choice, including consent withdrawal, please see our Privacy Policy.

Reviewer #1: Yes: Mohammad Sayyar Khan

Reviewer #2: Yes: Nii Korley Kortei

---

## [Author Response · Author response to Decision Letter 0]

20 Sep 2024

The responses to the comments are given below.

Reviewer #1:. 

1)Line 14- 15. The endophyte P. maximus is not included in the 12 isolates, then why it was shown as the most effective inhibitor of grapevine pathogen in the manuscript. This strain was not shown in either the abstract or in the tables.

Author response:-

Table (1) presents the identification of twelve endophytic fungi based on the analysis of ITS (Internal Transcribed Spacer) and SSU (Small Subunit) sequences. Other endophytes, including P. maximus, A. alternata, Penicillium radiatolobatum, and C. buchloes, were identified solely through ITS sequencing. These endophytic fungi were employed as biological agents against plant pathogenic fungi.

To address the reviewer's concern, there are two options: either to add P. maximosus to the abstract and relevant tables or to remove it from the biocontrol analysis. I would appreciate the reviewer's guidance on which approach would be most suitable

2)Some pathogenic fungi causing plant diseases were also isolated such as B.dothidea, and F.oxysporum. How these can be used as the biocontrol agents?

Author response:-

Although B. dothidea and F. oxysporum are primarily known as plant pathogens, their potential as biocontrol agents underscores the complexity of fungal interactions within plant ecosystems. Non-pathogenic strains of Fusarium oxysporum and B. dothidea have demonstrated biocontrol potential against various plant diseases in previous studies. For instance, Larkin and Fravel (2002) showed that a non-pathogenic strain of F. oxysporum effectively controlled Verticillium wilt in potato plants. Additionally, Meenu Katoch et al. (2014) reported that B. dothidea, when isolated as an endophytic fungus, can serve as a potential biocontrol agent.

3)Line 45-48. Please provide more citations explaining the endophytic antagonistic effects against B.ceneria and I.destructans, two of the most destructive pathogens of grapevine.

Author response:-

Provided more citations explaining the endophytic antagonistic effects against B.ceneria and I.destructans, two of the most destructive pathogens of grapevine.

4)Lines# 96-107. Provide details and quantities of PCR reaction mixture for both ITS and SSU. 

Author response:-

quantities of PCR reaction mixture for both ITS and SSU included: 

12.5 µl of Taq PCR master mix (Add Bio, Korea), 1 µl of both forward and revers primer of 10 pmol, 2 µl genomic DNA, and 8.5 µl RNase free water in 25µl of final volume reaction.

5)Several grammatical errors were found in lines # 116- 159.

Author response:-

Corrected 

6)Table 1 needs formatting. It’s not the correct style.

Author response:-

Corrected 

7)Grammatical errors in lines# 170- 185.

corrected

8)Fig. 1. Species names must be in italics. Provide reference to the NJ method.

Author response:-

Provided 

9)Line#207. Where is P.maximus in Table 1? 

Author response:-As responded in a comment (1).

"This study is part of a PhD project. P. maximus, A. alternata, and C. buchloes were identified only through ITS sequencing; therefore, they are not included in the table and are not mentioned in this manuscript. This manuscript (in the molecular study) includes only the fungi identified based on both ITS and SSU primers. 

The table below shows the fungi identified based on ITS primers only and not included in this study (in the Molecular identification section). 

Scientific name Per identity Accession number 

Paecilomyces maximus

100% PP410469

Alternaria alternata 99.81% PP410473

Curvularia buchloes

100.00% PP410464

Penicilium radiatolobatum 99.825 PP410466

10)Table 3. Not the correct style. Needs formatting.

Author response:-

Corrected

11)Fig 2 needs major revision. No horizontal and vertical titles, no error bars.

Author response:-

Corrected 

12)The results and findings have not been discussed properly in the light of previous literature. Only four references were mentioned to support the results of the present study.

corrected

Reviewer #2: 

Abstract

1. Line 2-5- The geographical location of the study location must be known eg. Nigeria etc

Author response:-

Added >>>> It is "Duhok Province in the Kurdistan Region of Iraq"

2. A brief introduction to the subject must be given in 2 lines

Author response:-

Added 

3. Line 9- " These fungi were "Intro

Author response:-

corrected

4. Line 51- Our study sought to....."

Author response:-

corrected

Methods

5. More than 1 medium should have been used to plate the samples since it creates more diversity in species of fungi. A better picture is given… 

Author response:-

Two media were used in our study: Potato Dextrose Agar (PDA) and Malt Extract Agar (MEA). As suggested by the reviewer, we included both media in our methodology. We did not initially mention MEA in the method section because the same fungi were isolated from both PDA and MEA.

6. Results from this present study must be compared with similarly one works from across the globe to ascertain differences and possible sound scientific reasons given.

Author response:-

corrected

7. PLOS authors have the option to publish the peer review history of their article (what does this mean?). If published, this will include your full peer review and any attached files.

Author response:- Yes

---

## [Decision Letter · Decision Letter 1]

30 Sep 2024

Molecular Identification and Antimicrobial Potential of Endophytic Fungi Against Some Grapevine Pathogens

PONE-D-24-32835R1

Dear Dr. Haleem,

We’re pleased to inform you that your manuscript has been judged scientifically suitable for publication and will be formally accepted for publication once it meets all outstanding technical requirements.

Kind regards,

Estibaliz Sansinenea

Academic Editor

PLOS ONE

Additional Editor Comments (optional):

Reviewers' comments:

Reviewer's Responses to Questions

**Comments to the Author**

1. If the authors have adequately addressed your comments raised in a previous round of review and you feel that this manuscript is now acceptable for publication, you may indicate that here to bypass the “Comments to the Author” section, enter your conflict of interest statement in the “Confidential to Editor” section, and submit your "Accept" recommendation.

Reviewer #1: All comments have been addressed

2. Is the manuscript technically sound, and do the data support the conclusions?

Reviewer #1: Yes

3. Has the statistical analysis been performed appropriately and rigorously? 

Reviewer #1: Yes

4. Have the authors made all data underlying the findings in their manuscript fully available?

Reviewer #1: Yes

5. Is the manuscript presented in an intelligible fashion and written in standard English?

Reviewer #1: Yes

6. Review Comments to the Author

Reviewer #1: (No Response)

7. PLOS authors have the option to publish the peer review history of their article (what does this mean?). If published, this will include your full peer review and any attached files.

Reviewer #1: **Yes: **Mohammad Sayyar Khan

---

## [Editor Report · Acceptance letter]

15 Oct 2024

PONE-D-24-32835R1 

PLOS ONE

Dear Dr. Haleem, 

I'm pleased to inform you that your manuscript has been deemed suitable for publication in PLOS ONE. Congratulations! Your manuscript is now being handed over to our production team.

Kind regards, 

on behalf of

Dr. Estibaliz Sansinenea 

Academic Editor

PLOS ONE